# Solving Tea Blending Problems Using Interactive Fuzzy Multi-Objective Linear Programming

**Saran Jarernsuk and Busaba Phruksaphanrat ***

Research Unit in Industrial Statistics and Operational Research, Industrial Engineering Department,
Faculty of Engineering, Thammasat School of Engineering, Thammasat University,
Pathum Thani 12121, Thailand
* Correspondence: lbusaba@engr.tu.ac.th

**Abstract:** Blending is a classical and well-known optimization problem that has been applied in the food, steel, and composite material industries. However, tea blending is more complicated than general problems due to the variety of products, processes, and sources of raw materials and semi-products. So, in this research, a fuzzy multi-objective model for the tea blending problem was proposed to minimize the total production cost and the deviation of quality target score; it provides a more robust and flexible method than existing models for complex real-world problems. Existing research works of a blending problem consider only raw material cost, but semi-product cost and processing cost are included in the proposed model that matches the actual case. Losses that occur during production are also incorporated. The selection of appropriate raw materials and semi-product sources can be obtained with the preferred levels of cost and quality by the proposed algorithm. The interactive fuzzy multi-objective programming to solve the problem has advantages over existing interactive programming methods. It is easy to manipulate interactively to obtain more efficient solutions than existing methods and both balanced and unbalanced solutions can be selected. The comparison of the results of an existing approach and the interactive fuzzy multi-objective programming algorithm for the tea industry is illustrated.

**Keywords:** blending problem; tea industry; interactive fuzzy programming; multi-objective decision making; fuzzy-efficient solution





## 1. Introduction

One of the most popular beverages in the world is tea. The diversity in tea products and complacency of consumers vary by region, age, etc. [1]. There are varieties of tea based on production techniques and raw materials used. The most important characteristics that differentiate types of tea are color, thickness, softness, and flavor [2,3]. Tea is now trendy in Thailand, especially tea mixed drinks. Types of tea primarily used for mixing are black tea, oolong tea, and green tea. Following the trend, tea production is increasing to meet the higher demand for tea.

The tea manufacturing process mainly involves various operations: withering, rolling, fermentation, roasting, and blending, depending on the type of product formula. All tea products are primarily manufactured by the same processes, though a few processes are different for each type of product. For example, black tea is fully fermented, while oolong tea is partially fermented. A withering process removes moisture to prepare leaves for the rolling process. In this process, partial oxidation occurs and is caused by disrupting the cell wall matrix (partially fermented) [4]. The rolling process is to mash the tea leaves into small pieces and allow mixing of polyphenol oxidase enzymes and polyphenols, which are necessary for oxidation [5]. For black tea products, fermentation is the complete oxidation process of the tannin content in the tea shoots caused by the action of enzymes in the leaf. As a result, tea shoots, which are colorless in their raw state, turn reddish-brown [6]. Finally,

the roasting process reduces the water content in the tea body, which causes a change in the taste and consistency of the brewed tea.

The motivation for this research arose from the case study of a tea company in Thailand that produces many types of tea. It is facing the problem of production planning for blending tea. Multi-process and multi-source production makes it difficult for the company to reach its goals. The issue of how to blend varieties of raw materials and how to order semi-products (e.g., fermented leaves) from outsourcing companies to satisfy the demand and quality of customers with specific characteristics and resources at the lowest costs has become a complicated task for the company. Moreover, each tea production process has different yields, making it problematic to estimate the number of finished products.

The tea blending problem is an optimization problem that usually tries to minimize the total cost with the availability of resources, the satisfactory condition of products, and the necessary processes. Each type of tea has different characteristics depending on the quality of raw materials and semi-products, so targeted characteristic scores of the final blend are typically set. Unfortunately, these targeted typical scores are difficult to reach due to the recipe of blending and processes. Therefore, tea production intends to find the closest solution that matches the characteristic targets at the lowest cost.

This research aims to construct and solve a tea blending model that can find the lowest total cost and determine the closest characteristics of the final products with the complicated constraints of the tea blending problem. Losses during the flow of materials are also considered. Moreover, an algorithm for interactive fuzzy multi-objective programming with priority control is presented as a tradeoff between the total cost and the quality score for the decision-maker (DM) to select. The model was tested for a tea company in Thailand. It was also compared with the existing interactive fuzzy programming model currently used to present the effectiveness of the proposed interactive fuzzy multi-objective programming algorithm.

The significant contributions of this research to the industry are reducing the cost and time for planning and increasing product quality by minimizing the deviations from the target characteristic scores. Moreover, the interactive fuzzy programming algorithm can improve the capability to handle the complex problem of tea blending with various products, processes, and sources. The interactive programming model allows the flexibility to evaluate the tradeoff between conflicting objectives. The academic contribution is an interactive fuzzy multi-objective programming algorithm that can obtain various efficient solutions, and both balanced and unbalanced solutions can be selected. The preferred solutions can be chosen interactively based on the DM's preference. The existing interactive fuzzy multi-objective programming method was compared. It was found that the proposed algorithm for the interactive fuzzy programming with priority control was superior to existing methods. It is easy to generate a satisfactory solution; there were various efficient solutions for DM to select, and both balanced and unbalanced solutions can be indicated and selected. To the best of our knowledge, there has not been any interactive fuzzy mathematical model applied to the tea blending problem previously.

This paper is organized as follows: Section 2 provides a review of the significant literature, Section 3 shows the interactive fuzzy programming with priority control mathematical model and solution method, and a practical real case study of a tea company is presented in Section 4. Finally, the conclusion is aggregated in Section 5.

## 2. Literature Review

The blending problem is a classical problem. The methods of solving this problem aim to achieve the specific blending needs for each product. There are many different applications in industries related to these kinds of issues. For example, in the steel casting and laminated composite industries, the blending problem is mainly concerned with selecting raw materials and their quantities to be blended [7–12]. This kind of issue is also similar to the situation in the food industry [2,13].

Existing models for blending problems concern the cost objective function. However, only raw material cost is included in the models [2,7,10–15]. Semi-product and process costs are also substantial and may vary depending on the sources of materials. In some factories, semi-products are provided by outsourcing companies, and each process has different costs that vary based on the type of products. The quality of products is also crucial for customers; only a few research works are included. Multi-process is the actual condition of tea production, which is rarely considered. This kind of problem is a network flow problem in which a loss occurs during processing.

Many methods have been applied for a single objective blending problem, such as mathematical programming [10,13,14,16,17], simulation [2,7], and genetic algorithm [9,11,12,15,18]. Both mathematical models and stochastic models were generally applied to the problem. General mathematical models examined demand, supply, and blending conditions [9,12,19]. For a multi-objective problem, goal programming approaches were used in the food and coal industries [14,16,17]. Costs and quality were the main objectives of the food industry. A coal blending problem was proposed to manage a consistent feed of blended coal to meet environmental and boiler performance requirements [17]. A Monte Carlo simulation approach was also employed to manage the multi-objective tea blending problem by randomly generating the weighted values of objectives [2]. Minimization of raw material cost and a violation of the target score were included, but multi-process and multi-source of raw materials were not considered. Tradeoff solutions could be obtained, but efficiency could not be guaranteed. Both cost and quality are the main concerns of the tea blending problem, so a multi-objective model should be applied.

Earlier, Zimmermann proposed fuzzy multi-objective linear programming to manage uncertainty [20]. Fuzzy goal programming was also presented for DM with specific knowledge of the different goals [21]. Afterward, interactive fuzzy programming approaches were proposed to find more efficient solutions to DM's preferences that made it more flexible for DM than existing fuzzy multi-objective methods. Moreover, tradeoff solutions among objectives could be obtained easily. Augmented max–min fuzzy approach by Lai and Hwang (1993) [22], interactive fuzzy programming approach by Torabi and Hassini (2008) [23], and extended Werner's fuzzy approach by Selim and Ozkarahan (2008) [24] are interactive fuzzy programming approaches that have been utilized in many studies. However, they still have some limitations in obtaining efficient solutions by assigning weighted values.

So, this research proposes an interactive fuzzy programming algorithm for the multi-objective tea blending problem with multi-product, multi-process, and multi-source of raw materials. The total production cost and the total deviations from the quality target score are minimized to find the best production plan. An interactive fuzzy programming method was adopted to transform the multi-objective linear programming model into an auxiliary crisp multi-objective, mixed-integer linear programming model. It is used to find the preferred solution by increasing the flexibility of multi-objective decision-making techniques to obtain efficient solutions. The classification of related literature on blending problems is summarized in Table 1.

**Table 1.** Review of the related studies of the blending problem.

| Authors | Case Study | Objective/ Type | Cost Objective | | | Quality Objective | Yield | Process/ Product | Solution Method |
|---|---|---|---|---|---|---|---|---|---|
| | | | Raw Material | Semi- Product | Process Normal/Express | | | | |
| (Steuer, 1984) [16] | Sausage | Multiple/ deterministic | y | n | n | y | y | s/s | Goal programming |
| (Olson, 1993) [13] | Sausage | Multiple/ deterministic | y | n | n | n | y | s/s | Interactive linear programming |
| (Lyu et al., 1995) [17] | Coal | Multiple/ deterministic | y | n | n | n | n | s/s | Goal programming |
| (Toklu, 2005) [15] | - | Multiple/ Stochastic | y | n | n | y | n | s/s | Genetic algorithm |
| (Xi-Jin et al., 2009) [11] | Coal | Single/ Stochastic | y | n | n | n | y | s/s | Genetic algorithm |
| (Fröhling and Rentz, 2010) [8] | Iron and steel | Single/ Stochastic | y | n | n | n | n | s/s | SCOPE Simulation |
| (Babić and Perić, 2011) [14] | Livestock feed blend | Multiple/ deterministic | y | n | n | n | n | s/s | Goal programming |
| (Sakall and Baykoç, 2011) [10] | Brass | Single/ Stochastic | y | n | n | n | y | s/s | Possibilistic programming |
| (Djeumou Fomeni, 2018) [2] | Tea | Multiple/ Stochastic | y | n | n | y | n | s/m | Monte Carlo simulation |
| (Chen et al., 2020) [7] | Zinc | Single/ Stochastic | y | n | n | n | n | s/s | Monte Carlo simulation and Genetic algorithm |
| (Yuan et al., 2020) [12] | Coal | Single/ Stochastic | y | n | n | n | y | s/s | Particle swarm optimization |
| (Ntourmas et al., 2021) [9] | Laminated composite | Single/ Stochastic | n | n | n | y | y | s/s | Genetic algorithm |
| (Haonan et al., 2021) [18] | Mining | Multiple/ Stochastic | n | n | n | y | n | m/m | Rolling-horizon heuristic |
| Proposed model | Tea | Multiple/ fuzzy | y | y | y | y | y | m/m | Interactive fuzzy programming |

Note: y = yes, n = no, m = multiple, s = single.

### 3. Proposed Mathematical Model

In tea production, the raw materials are obtained from tea fields and semi-products may be outsourced from other companies. The yield of each process affects the decisions of production planning and outsourcing quantity. After mixing both in-process and out-sourced materials, the quality and cost objectives of the products should be satisfied. This section presents a mathematical programming model for the blending problem in the tea industry to obtain the appropriate blending formulation and capacity allocation for multi-product, multi-process, and multi-source of materials.

Notations and Indices
$i$: set of product ($i = 1, \ldots, I$).
$j$: set of process ($j = 1, \ldots, J$).
$k$: set of raw material from supplier ($k = 1, \ldots, K$).
$s$: set of outsourcing company ($s = 1, \ldots, S$).
$c$: set of characteristic type ($c = 1, ..., C$).

Parameters
$d_i$: total demand for product $i$.
$y_j$: yield in process $j$.
$mr_k$: maximum amount of raw material from supplier $k$.
$mp_j$: maximum regular capacity of process $j$.
$mo_j$: maximum overtime capacity of process $j$.
$ms_{ijs}$: maximum outsourcing capacity of product $i$ in process $j$ from outsourcing $s$.
$c_{jk}$: cost per unit of regular production in process $j$ of raw material from supplier $k$.
$oc_{jk}$: cost per unit of overtime production in process $j$ of raw material from supplier $k$.
$sc_{ijs}$: cost per unit of product $i$ for process $j$ from outsourcing $s$.
$q_{ikc}$: characteristics score for product $i$ from the raw material of supplier $k$ with characteristic type $c$.
$qo_{isc}$: characteristics score for product $i$ from outsourcing $s$ with characteristic type $c$.
$r_{ic}$: score requirement for a product $i$ of characteristic $c$.

Decision variables
$P_{ijk}$: amount of product $i$ by regular production in process $j$ from raw material $k$.
$OP_{ijk}$: amount of product $i$ by overtime production in process $j$ from raw material $k$.
$OS_{ijs}$: amount of product $i$ in process $j$ from outsourcing $s$.
$Ne_{ic}$: negative deviation for product $i$ of characteristic type $c$.
$Po_{ic}$: positive deviation for product $i$ of characteristic type $c$.

### 3.1. Mixed Integer Linear Programming Model for Tea Production

The objective is to seek the minimum total production cost, as shown in (1), and to minimize the total deviation from the quality target score, as shown in (2). The total cost calculates from the price of fresh tea from each supplier, the cost of fermented semi-product from each outsourcing company, and the cost of each process for both regular and overtime productions. Customer satisfaction depends on product quality. The target quality of each characteristic of tea is set. The goal function is presented to minimize deviations from the target values, indicating the satisfactory quality of products [25]. The formulation of the model can be shown as follows:

$$\text{Minimize} \sum_{i=1}^{I} \sum_{j=1}^{J} \sum_{k=1}^{K} \left( (P_{ijk} * c_{ik}) + (OP_{ijk} * oc_{ik}) \right) + \sum_{i=1}^{I} \sum_{j=1}^{J} \sum_{s=1}^{S} (OS_{ijs} * sc_{is}) \tag{1}$$

$$\text{Minimize} \sum_{i=1}^{I} \sum_{c=1}^{C} (Ne_{ic} + Po_{ic}) \tag{2}$$

Constraints

$$\sum_{k=1}^{K} (P_{iJk} + OP_{iJk}) + \sum_{k=1}^{K} OS_{iJk} \geq d_i \quad \text{for all } i \tag{3}$$

$$P_{ij+1,k} = P_{ijk} * y_i \quad \text{for all } i, j \text{ and } k \tag{4}$$

$$OP_{ij+1,k} = OP_{ijk} * y_i \quad \text{for all } i, j \text{ and } k \tag{5}$$

$$OS_{ij+1,s} = OS_{ijs} * y_i \quad \text{for all } i, j \text{ and } s \tag{6}$$

$$\sum_{i=1}^{I} P_{i1k} \leq mr_k \quad \text{for all } k \tag{7}$$

$$\sum_{i=1}^{I} OS_{ijs} \leq ms_{ijs} \quad \text{for all } j \text{ and } s \tag{8}$$

$$\sum_{i=1}^{I} \sum_{k=1}^{K} P_{ijk} \leq mp_j \quad \text{for all } j \tag{9}$$

$$\sum_{i=1}^{I} \sum_{k=1}^{K} OP_{ijk} \leq mo_j \quad \text{for all } j \tag{10}$$

$$(( \sum_{k=1}^{K} (q_{ikc} * (P_{iJk} + OP_{iJk})) + \sum_{s=1}^{S} (qo_{isc} * OS_{iJs})) / ( \sum_{k=1}^{K} (P_{iJk} + OP_{iJk}) + \sum_{s=1}^{S} (OS_{iJs}))) + Ne_{ic} - Po_{ic} = r_{ic} \tag{11}$$
for all $i$ and $c$

$$P_{ijk}, OP_{ijk}, OS_{ijs}, Ne_{ic}, Po_{ic} \geq 0 \quad \text{for all } i, j, k, s \text{ and } c \tag{12}$$

The demand for each product should be satisfied, as shown in Equation (3). Equations (4)–(6) show the flow balance between consecutive processes. Supply and outsourcing capacities are shown in (7) and (8). The regular and overtime capabilities of the process are presented in (9) and (10). Equation (11) shows the goal constraints of characteristic requirements, and Equation (12) shows the non-negativity conditions. The tradeoff between the total cost and the product quality is an imprecise decision, so an interactive fuzzy programming model is applied.

### 3.2. Interactive Fuzzy Programming Methods

An interactive fuzzy programming method can convert the multi-objective model into an auxiliary crisp model based on a fuzzy decision. Existing interactive fuzzy methods are credited to Lai and Hwang [22], called the LH fuzzy approach, Selim and Ozkarahan [24], called the SO fuzzy approach, and Torabi and Hassini [23], called the TH fuzzy approach. The TH fuzzy approach has more advantages than the others in obtaining balanced or unbalanced compromise solutions. Moreover, it can obtain the compromise solution more sensitively than the SO fuzzy approach. The TH fuzzy approach applies a coefficient of compensation and the weight of either objective to manage balance and find efficient solutions. The model can be illustrated by Equation (13).

$$\max \gamma \lambda_0 + (1 - \gamma) \sum_i \theta_i \mu_i(x) \tag{13}$$

$$\text{s.t. } \lambda_0 \leq \mu_i(x) \; i = 1, \dots, n$$

$$\sum_i \theta_i = 1$$

$$x \in f(x) \text{ and } \lambda_0 \in [0, 1]$$

where $\lambda_0$ denotes a minimum satisfaction level of all objective functions; $\mu_i(x)$ is a membership function of $i$th objective; $f(x)$ is an objective function in the solution space of $x$; and $n$, $\gamma$, and $\theta_i$ are the number of objectives, the weighted values for controlling the minimum degree of satisfaction of all objectives, and a weighted value of the $i$th objective function based on the DM's preferences, respectively.

### 3.3. Interactive Fuzzy Programming with a Priority Control Method

Existing interactive fuzzy approaches can solve efficient solutions for the DM. However, there is a weakness because limited solutions are obtained. This research uses interactive fuzzy programming with the priority control method by Jarernsuk and Phruksaphanrat [26] to apply to the tea blending problem. The model uses the last priority objective and the compensation coefficient to control the number of efficient solutions instead of weight values [26]. This approach consists of max–min and compensatory terms with a priority constraint, making the model more effective and adaptable than existing interactive fuzzy programming methods. The model is illustrated in Equation (14).

$$\max \lambda = \gamma \lambda_0 + (1 - \gamma)( \lambda_1) \tag{14}$$

$$\text{s.t. } \lambda_0 \leq \mu_i(x) \ i = 1, \ldots, n$$

$$\mu_1(x) \geq \lambda_1$$

$$\mu_n(x) \geq a_n^*$$

$$x \in f(x), \lambda_0, a_n^* \in [0, 1], \text{ and } \gamma \in (0,1)$$

$\lambda_0$ and $\lambda_1$ denote a minimum satisfaction level of all objective functions and a minimum satisfaction level of the first priority objective; $\mu_i(x), \mu_1(x)$, and $\mu_n(x)$ are the membership functions of $i$th objective, the membership function of the objective $i$, first objective, and the lowest objective priority, respectively; $\gamma$ is the coefficient of compensation for all objectives; and $a_n^*$ is a minimum satisfaction level of the last priority objective that can be adjusted. This model can find more efficient solutions than existing methods.

### 3.4. Solution Procedure of Improving the Interactive Fuzzy Programming

Adjusting the values to select various solutions is not easy. Therefore, an initial step should be performed before changing the value of the parameters to find answers. This research presents an algorithm that considers the balanced solution initially. Then, appropriate solutions from the balanced solution to the first priority are generated that can reduce the scope of the solutions. The balanced solution can be obtained by configuring large values of the compensation coefficient ($\gamma = 0.99$) so that the objective equation is respected to a minimum satisfaction level of all objective functions [26]. In addition, the minimum satisfaction level of the last priority objective should be initially adjusted to 0.5 to reflect the balance of consideration. Then, a balanced solution and the value of the lowest priority membership function, $\mu_n^*(x)$ are obtained. This value of the membership function is used to set as the upper bound for $a_n^*$. Afterward, the DM can find efficient solutions by adjusting the value of $a_n^*$ and $\gamma$. The solutions are limited from a balanced solution to the first priority solution.

The solution procedure for improving the interactive fuzzy programming with the priority control method can be aggregated in the following steps [26]:

Step 1: Solve each objective (Equations (1) and (2)) under the problem constraints (Equations (3)–(12)) to find a positive ideal solution (PIS), $f_i^*$, and a negative ideal solution (NIS), $f_i^-$ [23]. For this problem, $f_1(x)$ is to minimize total production cost and $f_2(x)$ is to minimize the total deviations from the quality target score.

Step 2: Construct the membership functions of objectives by Equation (15):

$$(f_i(x)) \begin{cases} 1 & if \ f_i(x) < f_i^* \\ 1 - \frac{f_i(x) - f_i^*}{f_i^- - f_i^*} & if \ f_i^* \leq f_i(x) \leq f_i^- \\ 0 & if \ f_i(x) > f_i^- \end{cases} \tag{15}$$

Step 3: After transforming the objective functions into membership functions, the interactive fuzzy programming with a priority control model can be applied as Equation (14). Next, define the minimum satisfaction level ($a_n^*$) as 0.5 and the coefficient of compensation ($\gamma$) equal to 0.99 to find the balanced solution. Then, the membership function of the

lowest priority objective, $\mu_n^*(x)$, is obtained. If the balance solution is satisfactory, then stop. Otherwise, continue with step 4.

Step 4: Interactively adjust the value of $a_n^*$ between 0 to $\mu_n^*(x)$ (exclude 0) and adjust the value of $\gamma$ between zero to 0.99 (exclude 0) in model Equation (14). Then, solve for efficient solutions. The $\gamma$ value affects the balanced and unbalanced solutions, and $a_n^*$ affects the direction of the solution. For example, if a small value is assigned, the answer is directed to the first priority solution.

Step 5: If the DM is satisfied by the current efficient solution, then stop; otherwise, solve for other efficient solutions by changing the values of $\gamma$ and $a_n^*$ until the solution is satisfied.

## 4. A Case Study of a Tea Company

A case study of a tea company in Thailand was undertaken to illustrate the effectiveness of the proposed model. In this case study, there are two types of products, oolong tea (product 1) and black tea (product 2). Demands are rolling forecast quarterly. Some production processes of these teas are the same and some processes are different. In total, five processes for each product, starting from buying fresh tea from suppliers, a withering process (transform fresh tea to dry tea), a rolling process, a roasting process for oolong tea product, a fermenting process for black tea product, and a blending process, as shown in Figure 1. Firstly, the company has six suppliers of fresh tea having different characteristic scores of quality and prices. Color, coarseness, and softness are three characteristics of the raw material that the factory is concerned. Fresh tea is transformed into dry tea by the withering process. Next, the dry tea is rolled so that enzymes in the tea leaves are released in the rolling process. Different types of tea have other techniques for roasting and fermenting. Tea leaves in the roasting process have a slightly yellow to brown color and a scented aroma. The roasting process is used for oolong tea products. Other leaves are fully fermented in the fermentation process. Two outsourcing companies supply the fermenting process' semi-products (e.g., fermented leaves). Thus, the supplies have different characteristic scores of quality and prices. Tea leaves from the fermenting process become dark red with a fermented aroma and are used for black tea products. In the final approach, each product is dried and blended. The algorithm for the interactive fuzzy programming with a priority control method was applied to this practical problem of the tea company. Parameters are illustrated in Tables 2–5.

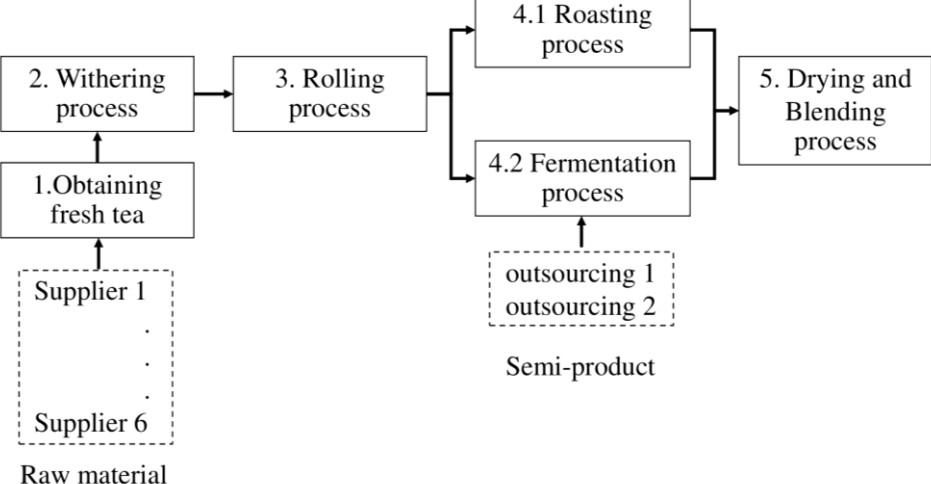

**Figure 1.** Flow process diagram of the tea blending factory.

**Table 2.** Parameters of the case study.

| No. | Parameter | Values | No. | Parameter | Values |
|---|---|---|---|---|---|
| 1 | $d_1$ | 8800 kgs | 21 | $c_{15}$ | 75 baht per kg |
| 2 | $d_2$ | 12,000 kgs | 22 | $c_{16}$ | 90 baht per kg |
| 3 | $mr_1, mr_2$ | 20,000 kgs | 23 | $c_{21}, \ldots, c_{26}$ | 10 baht per kg |
| 4 | $mr_3, \ldots, mr_6$ | 30,000 kgs | 24 | $oc_{21}, \ldots, oc_{26}$ | 15 baht per kg |
| 5 | $mp_2$ | 40,000 kgs | 25 | $c_{31}, \ldots, c_{36}$ | 8 baht per kg |
| 6 | $mo_2$ | 10,000 kgs | 26 | $oc_{31}, \ldots, oc_{36}$ | 12 baht per kg |
| 7 | $mp_3$ | 40,000 kgs | 27 | $c_{41}, \ldots, c_{46}$ | 9 baht per kg |
| 8 | $mo_3$ | 10,000 kgs | 28 | $oc_{41}, \ldots, oc_{46}$ | 12 baht per kg |
| 9 | $mp_4$ | 25,000 kgs | 29 | $c_{51}, \ldots, c_{56}$ | 12 baht per kg |
| 10 | $mo_4$ | 7500 kgs | 30 | $oc_{51}, \ldots, oc_{56}$ | 18 baht per kg |
| 11 | $mp_5$ | 20,000 kgs | 31 | $sc_{251}$ | 160 baht per kg |
| 12 | $mo_5$ | 5000 kgs | 32 | $sc_{252}$ | 180 baht per kg |
| 13 | $mp_6$ | 20,000 kgs | 33 | $c_{61}, \ldots, c_{66}$ | 15 baht per kg |
| 14 | $mo_6$ | 8000 kgs | 34 | $oc_{61}, \ldots, oc_{66}$ | 20 baht per kg |
| 15 | $ms_{251}$ | 20,000 kgs | 35 | $y_1$ | 1 |
| 16 | $ms_{252}$ | 20,000 kgs | 36 | $y_2$ | 0.4 |
| 17 | $c_{11}$ | 25 baht per kg | 37 | $y_3$ | 0.9 |
| 18 | $c_{12}$ | 45 baht per kg | 38 | $y_4$ | 0.8 |
| 19 | $c_{13}$ | 45 baht per kg | 39 | $y_5$ | 0.6 |
| 20 | $c_{14}$ | 75 baht per kg | 40 | $y_6$ | 0.8 |

**Table 3.** Scores for the raw material characteristics.

| Raw Materials | Characteristic Scores for Product 1 | | | Characteristic Scores for Product 2 | | |
|---|---|---|---|---|---|---|
| | Color (1) | Coarseness (2) | Softness (3) | Color (1) | Coarseness (2) | Softness (3) |
| 1 | $q_{111} = 8$ | $q_{112} = 2$ | $q_{113} = 8$ | $q_{211} = 1$ | $q_{212} = 9$ | $q_{213} = 2$ |
| 2 | $q_{121} = 1$ | $q_{122} = 3$ | $q_{123} = 7$ | $q_{221} = 3$ | $q_{222} = 8$ | $q_{223} = 3$ |
| 3 | $q_{131} = 6$ | $q_{132} = 8$ | $q_{133} = 2$ | $q_{231} = 7$ | $q_{232} = 3$ | $q_{233} = 4$ |
| 4 | $q_{141} = 2$ | $q_{142} = 5$ | $q_{143} = 6$ | $q_{241} = 4$ | $q_{242} = 4$ | $q_{243} = 5$ |
| 5 | $q_{151} = 5$ | $q_{152} = 7$ | $q_{153} = 3$ | $q_{251} = 3$ | $q_{252} = 6$ | $q_{253} = 4$ |
| 6 | $q_{161} = 4$ | $q_{162} = 5$ | $q_{163} = 5$ | $q_{261} = 6$ | $q_{262} = 5$ | $q_{263} = 5$ |

**Table 4.** Characteristic scores for semi-products from outsourcing companies.

| Outsourcing Companies | Characteristic Scores for Product 2 | | |
|---|---|---|---|
| | Color (1) | Coarseness (2) | Softness (3) |
| 1 | $qo_{111} = 4$ | $qo_{112} = 5$ | $qo_{113} = 5$ |
| 2 | $qo_{121} = 5$ | $qo_{122} = 5$ | $qo_{123} = 7$ |

**Table 5.** Target scores for each product.

| Target Scores of Product 1 | | | Target Scores of Product 2 | | |
|---|---|---|---|---|---|
| Color (1) | Coarseness (2) | Softness (3) | Color (1) | Coarseness (2) | Softness (3) |
| $r_{11} = 4$ | $r_{12} = 6$ | $r_{13} = 4$ | $r_{21} = 5$ | $r_{22} = 5$ | $r_{23} = 6$ |

Intel (R) Core (TM) i7-8550U CPU @ 1.80 GHz RAM 8.00 GB 64-bit and Lingo 17.0 software were used to solve the tea blending problem. From the interactive fuzzy programming with a priority control model, the PIS and NIS for this problem were $f^*(x) = (3{,}569{,}306, 0.857)$, $f^- = (4{,}457{,}910, 14.101)$, respectively. The best and the worst total cost were 3,569,306 and 4,457,910 baht, while the best and worst total deviations from

the quality target score were 0.857 and 14.101. Each objective function coincides with an equivalent membership function, which can be received using Equations (16) and (17):

$$\mu(f_1(x)) \begin{cases} 1 & if \ f_1(x) < 3,569,306 \\ 1 - \frac{f_1(x)-3,569,306}{4,457,910-3,569,306} & if \ 3,569,306 \leq f_1(x) \leq 4,457,910 \\ 0 & if \ f_1(x) > 4,457,910 \end{cases} \tag{16}$$

$$\mu(f_2(x)) \begin{cases} 1 & if \ f_2(x) < 0.857 \\ 1 - \frac{f_2(x)-0.857}{14.101-0.857} & if \ 0.857 \leq f_2(x) \leq 14.101 \\ 0 & if \ f_2(x) > 14.101 \end{cases} \tag{17}$$

The interactive fuzzy programming with a priority control method was applied and is shown in Equation (18).

$$\max \lambda = \gamma \lambda_0 + (1 - \gamma)(\lambda_1) \tag{18}$$

$$s.t. \ \mu_i(f_i(x)) \geq \lambda_0 \text{ for } i = 1, 2.$$

$$\mu_1(f_1(x)) \geq \lambda_1$$

$$\mu_2(f_2(x)) \geq a_2^*$$

$$x \in f_i(x), \ \lambda_0, a_2^* \in [0, 1], \text{ and } \gamma \in (0,1)$$

Table 6 compares the results of the TH fuzzy approach and the interactive fuzzy programming with a priority control method at $\gamma$ values of 0.1–0.9. The interactive fuzzy programming with a priority control method could calculate several solutions by adapting the degree of satisfaction of the last objective. The TH fuzzy approach could find solutions by adjusting the weighted value with a step increase of 0.1 for each objective but could give only four different efficient solutions. The interactive fuzzy programming with priority control method could produce nine different efficient solutions, which were far more than the TH fuzzy approach, as shown in Figures 2 and 3.

**Table 6.** Possible solutions of the problem by the TH fuzzy approach and the interactive fuzzy programming with priority control method.

| Interactive Fuzzy Programming with Priority Control ($a_2^*$ ($\Delta$0.1)) | | TH Fuzzy Approach ($w_1$, $w_2$ ($\Delta$0.1)) | |
|---|---|---|---|
| $f_1(x)$ | $f_2(x)$ | $f_1(x)$ | $f_2(x)$ |
| 3,954,113 | 2.181 | 3,954,113 | 2.181 |
| 3,789,629 | <u>4.14</u> | 3,789,629 | <u>4.14</u> |
| 3,714,676 | 5.033 | 3,714,676 | 5.033 |
| 3,569,306 | 9.163 | 3,569,306 | 9.163 |
| **3,842,932** | **3.505** | | |
| **3,731,751** | **4.83** | | |
| **3,675,212** | **6.154** | | |
| **3,628,588** | **7.479** | | |
| **3,581,964** | **8.803** | | |

Underlined solution is a balanced solution. Bold solutions are the different solutions.

The balanced solution was found when $\gamma$ was set to 0.99 and $a_2^*$ was varied from 0 to 0.5 (exclude 0). The solution was 3,789,629 baht for the first objective and 4.14 for the second objective at the value of $\mu_2^*(x) = 0.752$. Therefore, the value of $a_2^*$ was varied between 0 and 0.752 (exclude 0) in the model Equation (14). In this example, the value ranged from 0.1 to 0.75.

Table 7 shows the solutions for the proposed algorithm of interactive fuzzy programming with a priority control. There were six solutions, as shown in Table 7, while TH fuzzy acquired only two solutions (the balanced and the first priority solutions). This algorithm makes interactive fuzzy programming with a priority control more concise than the original one. As a result, it is easier for the DM to choose the preferred solution.

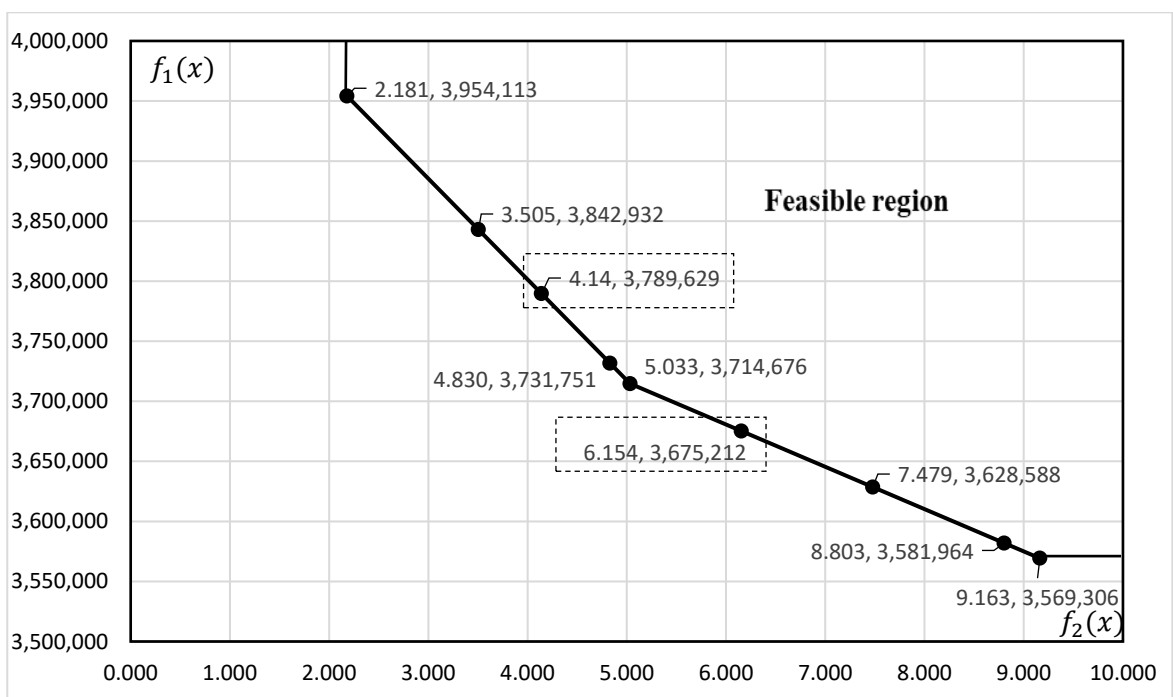

**Figure 2.** The result from the TH fuzzy approach.

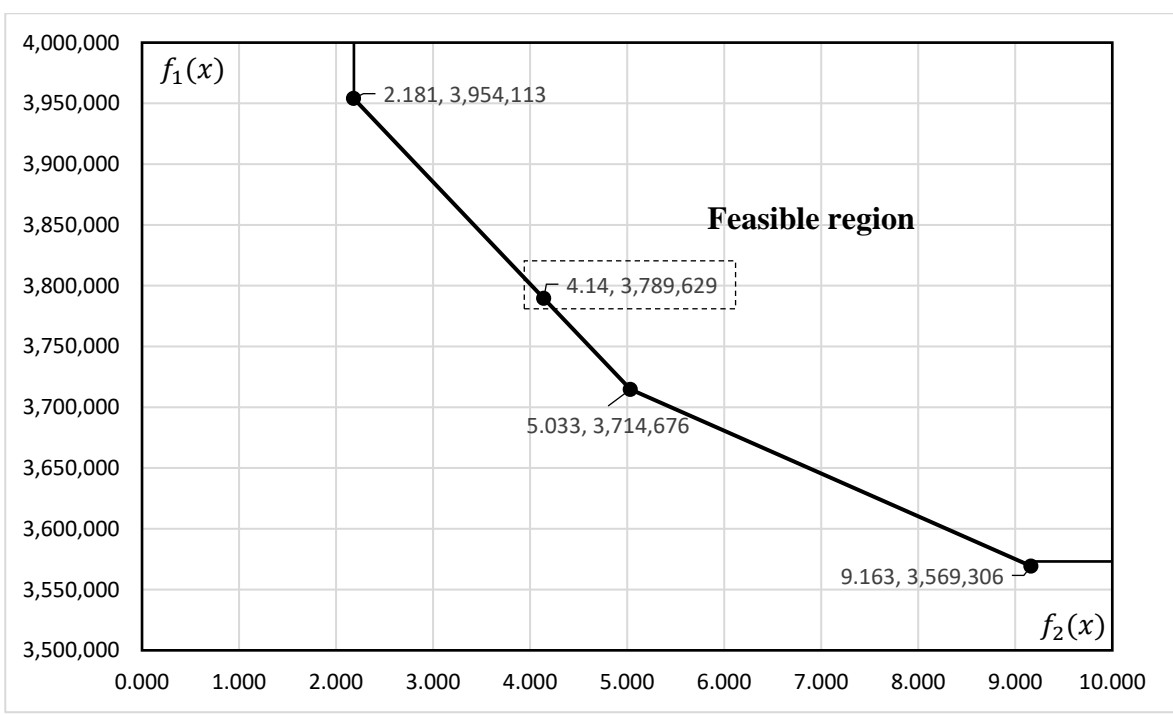

**Figure 3.** The result from the interactive fuzzy programming with a priority control method.

**Table 7.** Possible solutions to the problem by interactive fuzzy programming with a priority control algorithm.

| Interactive Fuzzy Programming with a Priority Control Algorithm at $\gamma = 0$ | | | | |
|---|---|---|---|---|
| $a_2^*$ | $\mu_1(x)$ | $\mu_2(x)$ | $f_1(x)$ | $f_2(x)$ |
| 0.1–0.3 | 1 | 0.37 | 3,569,306 | 9.163 |
| 0.4 | 0.98 | 0.4 | 3,581,964 | 8.803 |
| 0.5 | 0.93 | 0.5 | 3,628,588 | 7.479 |
| 0.6 | 0.88 | 0.6 | 3,675,212 | 6.154 |
| 0.7 | 0.82 | 0.7 | 3,731,751 | 4.83 |
| 0.752 | 0.752 | 0.752 | 3,789,629 | 4.14 |

An unbalanced solution may be considered; however, the satisfaction level of the lower priority objective should not be overrated because the satisfaction level of the first priority would be reduced. In this problem, the DM considered that the main objective that should be minimized would be the total production cost, so the degree of satisfaction with the second objective was set to 0.6 and the compensation coefficient was set to 0.1. The result for the total cost was 3,675,212 baht and the positive and negative deviation quality score was 6.154, which was an acceptable value for the DM, close enough to the target quality score. This total cost is lower than the balanced solution suggested by the TH fuzzy approach (3,789,629 baht). This unbalanced solution satisfied the DM that existing methods could not obtain it. Therefore, both the TH approach and interactive fuzzy programming with a priority control method can provide efficient solutions. However, the interactive fuzzy programming with a priority control method can produce more efficient solutions than the TH method, so the DM has more production and purchasing planning choices.

Djeumou Fomeni (2018) [2] proposed a method for the tea blending problem using a stochastic model for a single process. Their model uses randomly selected weights to combine objective functions. Therefore, only the extreme points in the case study were found. From the various values for the weights, there are only three solutions. Two of them were the best solution for a single objective, so there was one solution for the multi-objective problem ($f_1(x) = 3,569,306$ baht, $f_2(x) = 9.163$). The proposed method can find more diverse answers.

## 5. Conclusions

This research proposed a fuzzy multi-objective multi-product, multi-process, and multi-source tea blending model that aims to minimize the total production cost and the total deviation from the quality target score. Existing models consider only raw material costs but do not consider semi-product and processing costs that vary depending on the quality of raw materials and semi-products. It is an applicable model for the real case study in which losses occur in each production stage. Since the quality of the ingredients of the blends varies, the model tries to optimize and find a compromise solution that satisfies both objective functions by this interactive fuzzy programming with a priority control method. The conventional interactive fuzzy programming methods have a limitation in obtaining efficient solutions using weighted values. In contrast, interactive fuzzy programming with a priority control method uses the priority satisfaction level of the last priority objective to obtain many more feasible solutions. The DM may not be satisfied with the limited solutions, so the interactive fuzzy programming with priority control method can improve the feasible choices for multi-objective problems with a greater number of efficient solutions than existing methods. In addition, preferred solutions can be selected easily by adjusting the coefficient of compensation and the minimum degree of satisfaction of the lowest objective priority. The proposed algorithm for interactive fuzzy programming with a priority control in this research made it easy to define the solution range by generating solutions from a balanced solution to the best solution for the first priority objective. Existing methods cannot find this kind of solution.

In future research, a supply chain network for tea production would be interesting to investigate because it is a multi-stage process with losses at each stage, and time constraints in the network are also significant for the quality of products. The tea storage time and condition of the storage affect the quality of tea; conversely, holding costs will be increased. The proposed model can also be applied to other applications such as dairy production, food production, and beverage production. Furthermore, the proposed solution method requires DM to select the priority level for each objective. In some cases, it may not be easy to decide. Multi-criteria decision-making (MCDM) can be used for additional calculations to analyze appropriate ranks according to the criteria under consideration.

**Author Contributions:** Conceptualization, S.J. and B.P.; methodology, S.J. and B.P.; validation, S.J. and B.P.; formal analysis, S.J.; investigation, S.J.; resources, S.J. and B.P.; data curation, S.J. and B.P.; writing—original draft preparation, S.J.; writing—review and editing, B.P.; visualization, S.J.; supervision, B.P.; project administration, B.P.; funding acquisition, B.P. All authors have read and agreed to the published version of the manuscript.

**Funding:** This research was supported financially by research funding from the Faculty of Engineering, Thammasat School of Engineering, Thammasat University Research Unit in Industrial Statistics and Operational Research, Thammasat University, Thailand.

**Institutional Review Board Statement:** Not applicable.

**Informed Consent Statement:** Not applicable.

**Data Availability Statement:** Data is contained within the article.

**Conflicts of Interest:** The funders had no role in the design of the study; in the collection, analyses, or interpretation of data; in the writing of the manuscript, or in the decision to publish the results.

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
