# Peer review of "Solving Tea Blending Problems Using Interactive Fuzzy Multi-Objective Linear Programming"

_processes, doi:10.3390/pr11010049_

Round 1

Reviewer 1 Report

This work addressed the tea blending problem for a tea company in Thailand. The authors propose a fuzzy multi-objective multiproduct, multiprocess, and multisource tea blending model, aiming to minimize two objectives: the total production cost and the total deviation from the quality target score. Furthermore, they compare the existing interactive fuzzy multi-objective programming method and the proposed algorithm of the interactive fuzzy programming with priority control, being their approach superior and easy for generating satisfactory solutions for the DM.

The work's main contribution is the proposed algorithm of interactive fuzzy programming with priority control applied to a real case study. The work is interesting and, in general, well-written. Therefore, I believe the manuscript could be accepted if the following comments can be addressed:

1. On page 3, Section 2, lines 101-102, the last phrase should move to the end of this section.

2. On page 3, Section 2, line 112, these references should be associated with each topic like that: "genetic algorithm [9, 11, 16]".

3. On page 3. Section 2, line 131, the abbreviations TH, LH, and SO are strange. Then, understanding that it is for the authors' name. Anyway, fix explaining it or define abbreviations later.

4. On page 5, Section 3:

a) I suggest you define the sets instead of indices, like that, "I: Set of products, $i \in \{1, ..., |I|\}.

b) There are many ellipses with only two points. Fix them.

c) All the symbols should be italicized or use $$ with latex.

d) line160, indice k must be k = 1, ..., K, right?

5. On pages 5-6, Section 3.1, in the mixed integer linear programming model:

a) I suggest the summations should be with subscripts and superscripts instead of them on the right side to be clearer and more understandable.

b) In Eq (3), what happened with the j index? Is fixed, or is there a summation?

c) in each constraint of the model, change "for [index] = ..." to "for all" mathematics symbol.

d) Which variables are integers P, OP, and OS? Explicit them.

e) Ne is a negative deviation, so is it an unbounded variable?

f) In constraints (11), only P and OP have i, j, and k. Ne and Poic have i and c. Finally, OS has i, j, and s.

6. On page 6, Section 3.2, "… are credited to Lai and Hwang (1993), called LH fuzzy approach, Selim and Ozkarahan (2008), called SO fuzzy approach, and Torabi and Hassini (2008), called TH fuzzy approach. TH fuzzy …" → "… are credited to Lai and Hwang [23], called LH fuzzy approach, Selim and Ozkarahan [25], called SO fuzzy approach, and Torabi and Hassini [24], called TH fuzzy approach. TH fuzzy …".

7. On page 6, Section 3.2, Eqs (12) and (13)n, x, and f(x) are not defined.

8. On page 6, Section 3.3, "Jarernsuk and Phruksaphanrat (2022)" "Jarernsuk and Phruksaphanrat [27]".

9. On page 7, I suggest adding from line 250 a new subsection should be called "Solution procedure of improving the interactive fuzzy programming".

10. Future research is weak. Supply 3-5 solid and insightful future research suggestions in the new problem (in at least one separate paragraph) for the Processes community.

11. Minor issues:

a) In lines 290-291," In total, 5 processes for each", 5 → five.

b) In line 294, "company has 6 suppliers of fresh", 6 → six.

c) In line 307, the period is missing at the end of the phrase.

d) In line 402, "give only 4 different", 4 → four.

e) In line 403, "could produce 9 different", 9 → nine.

f) Table 6, problems with the brackets in the head columns.

g) Table 6, what mean * (asterisk)?

h) In line 461, "There were 6 solutions", 6 → six.

i) In line 462, "got only 2 solutions", 2 → two.

j) References: I suggest always using full journal names.

Reviewer 2 Report

The paper deals with the tea blending problem. A fuzzy multi-objective model that considers raw material cost, semi-product cost and processing cost was proposed. Experiments are done on a practical real case study of a tea company to evaluate the proposed model. The paper is generally well written, however, I encourage authors to submit a revised version of the paper along the comments in the reviewing report.

Round 2

Reviewer 2 Report

No comments